# VQ-SAD: Vector Quantized Structure Aware Diffusion For Molecule Generation

**Anonymous Author(s)** [1]

## Abstract

Many diffusion based molecule generation methods ignore the symbolic information of molecules and represent the atom and bond type as one hot representation. Methods based on Morgan fingerprints produce hash collisions and are hard to embed into a continuous space without information loss and random fingerprints correspond to no valid molecule. To circumvent this issue we use another paradigm and consider atom and bond codes as latent variables of VQ-VAE. We introduce VQ-SAD which first trains a VQ-VAE and uses the frozen pretrained VQ-VAE model and considers the codebooks for both atom and bond types as tokenizers for the downstream diffusion process. VQ-SAD is a neuro-symbolic model that utilizes both symbolic and neural structural information for a diffusion based model with learnable forward process. The large discrete code space provides a more balanced atom and bond types which enhances the denoising process. VQ-VAE slightly outperforms SOTA models for diffusion based molecule generation on QM9 and ZINC250k datasets.

## 1. Introduction

Diffusion based generative modeling of molecules has emerged as a central research area in machine learning, with applications spanning molecular discovery, protein design, and drug design. Symbolic information plays a central role in molecular structure, capturing discrete chemical concepts such as functional groups, valence rules, and reaction-relevant substructures that are difficult to encode using conventional one-hot atom and bond features. However, most diffusion-based molecular generative models like DiGress (Vignac et al., 2023), PARD (Luo et al., 2024) rely on categorical encodings, limiting their ability to leverage the rich symbolic chemistry underlying molecular graphs.

An important question is: **How could we encode symbolic structural information in molecular modelings in diffusion based molecule generation ?** There are two major paradigms to resolve this issue. The first paradigm is based on Morgan fingerprints (including ECFP variants) (Joshi & Guven, 2025) which are binary vectors that show presence or absence of substructures. Unfortunately different molecules can have identical fingerprints and this is known as collision (Virany & Tripp, 2025). Hash collisions are unavoidable since fingerprints are not bijective and reconstructing the original molecule from its fingerprint is not possible precisely. For example, a single atom change can flip many bits. The second paradigm is leveraging discrete latent spaces such as the latents in VQ-VAE. The VQ-VAE codebook becomes a learned vocabulary of recurring patterns and produces interpretable latent dimensions where each code corresponds to a symbolic entity. These discrete codes in the codebook allow symbolic mixing of fragments and indirectly distinguish a carbon as is shown in figure 2 which illustrates that the atom carbon in two molecules can have different contexts without explicitly enumerating them.

Modeling the intrinsic symbolic information in molecules bears another fruit which balances the atom and bond types since many real-world molecule datasets suffer from inherent sparsity of class types , where certain node or edge categories appear far less frequently than others. This imbalance can hinder the denoiser's ability to accurately reconstruct underrepresented types, leading to biased or incomplete graph generations. To address these limitations, VQ-SAD leverages vector-quantized variational autoencoders (VQ-VAEs) (van den Oord et al., 2017), (Razavi et al., 2019) which provides a promising solution by mapping continuous node and edge representations into a discrete, learned codebook of latent types. By effectively quantizing the graph attributes, VQ-SAD promote a more uniform distribution of node and edge types, allowing the denoiser to operate over a richer and more balanced set of categories, which in turn improves the quality and diversity of the generated graphs.

One aspect of modeling structural information could be resolved by symbolic discrete representations of each atom type and bond type. The other aspect is the neural representation which can be efficiently modelled by relative ran-

---

*Equal contribution [1]Anonymous Institution. **AUTHORERR: Missing \icmlcorrespondingauthor.**

*Proceedings of the 42nd International Conference on Machine Learning*, Vancouver, Canada. PMLR 267, 2025. Copyright 2025 by the author(s).

dom walk probabilities (RRWP). However, a problem arises: **How could the structural information adaptively affect the noise scheduling if the forward process in diffusion is learnable like in MELD (Seo et al., 2025) ?** Other than structural representation, there exists other design considerations such as paralizability, validity and learnable forward process in diffusion based graph generation as is shown in Figure 1. VQ-SAD enhances MELD by making more expressive noise scheduler that is structurally aware, and the forward diffusion is conditioned on structural information of individual nodes rather than just relying on node types. This additionally mitigates the risk of the state collapse problem discussed in MELD. VQ-SAD has the following three

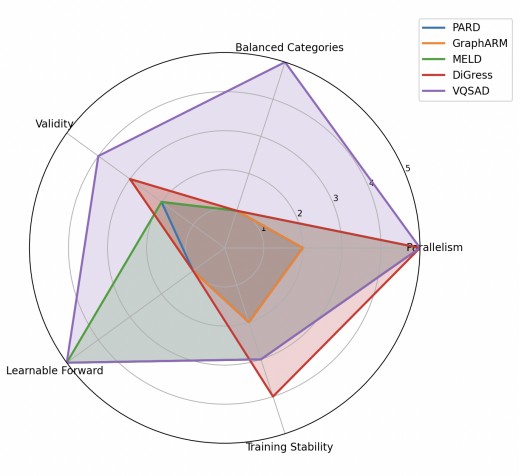

*Figure 1.* radar graph to compare VQ-SAD with other similar methods

contributions:

1. Developing a new methodology to leverage structural information in learning the forward process of diffusion based graph generation and conditioning the schedulars based on these structural features.

2. Considering VQ-VAE as a pretrained model and taking it as a fixed node and edge tokenizer and fixed decoder to have more balanced categories.

3. VQ-SAD achieves more efficient training and slightly outperforms SOTA models for diffusion based molecule generation for QM9 and ZINC250k datasets.

## 2. Related Work

### 2.1. Adaptive Noise Schedules

Several recent studies propose parametrizing the forward diffusion process and learning it jointly with the reverse process (Bartosh et al., 2024a), (Bartosh et al., 2024b) .

Learning the forward process provides several benefits such as better matching the data distribution so that the reverse process is easier to learn, and it allows the model to handle categorical distributions more effectively. For graph generation, (Kong et al., 2023) defines a diffusion ordering network that learns a data-dependent node absorbing ordering for the forward process in discrete graph space. (Seo et al., 2025) learns the forward process and needs gumble softmax to make it differentiable. Our work follows this adaptive noise scheduling but uses the structural information of the graph in learning these noise schedules for both node and edges.

### 2.2. VQ-VAE

(Yang et al., 2024) proposes a VQ-VAE style tokenizer to encode local graph substructures into discrete codes. The approach improves node and graph representation learning by capturing structural patterns while reducing computational cost.(Zeng et al., 2025) introduce a hierarchical VQ-VAE for graphs with a two-layer codebook to address codebook sparsity. This method enhances graph reconstruction and representation by learning richer, multi-level discrete embeddings for nodes and subgraphs. (Gu et al., 2022) leverages VQ-VAE for diffusion of images and encodes them into discrete tokens, and introduces a conditional diffusion model that operates in the latent token space. Our work is similar to this paradigm but is applied to nodes and edges of graphs. (Xia et al., 2023) uses a combination of several modules like context-aware tokenizer, masked-atom modeling (MAM), and contrastive graph-level learning. This combination of methods in (Xia et al., 2023) is computationally intensive compared to simpler approaches; to address this, we adopt diffusion-based methods for molecular learning.

## 3. Methodology

### 3.1. Motivation

Figure 2 shows that atom carbon in two molecules can have different contexts. The input in the denoiser is usually a one hot representation that collapses these two different types of carbons into one representation. We want to encode symbolic information before feeding it into the denoiser. We call this paradigm neurosymbolic approach that is shown in figure 3. The conventional GNN or transformers do not receive an informative representation since they only encode it by a one hot representation. GNNs often encounter oversmoothing issues (Gu et al., 2022), and increasing the number of message-passing layers can result in the loss of symbolic or structural information crucial for downstream tasks.

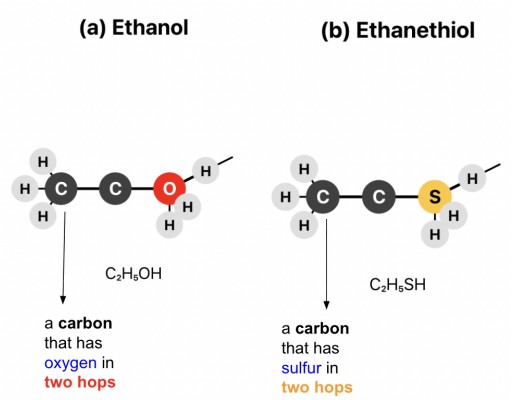

Figure 2. in (a) the carbon has a symbolic representation that indicates it is two hops away from oxygen. (b) illustrates a carbon atom that is two hops away from sulfur.

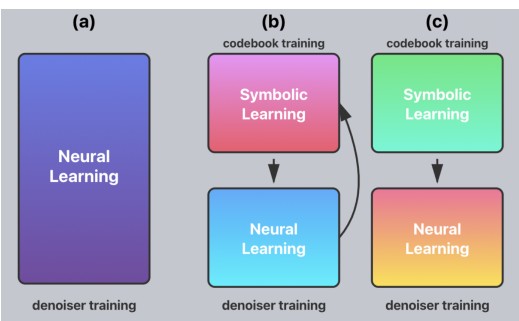

Figure 3. (a) Conventional denoiser that ignores symbolic context. (b) Learns both tokenizer and denoiser in parallel (unstable). (c) Proposed VQ-SAD: tokenizer frozen, denoiser receives discrete symbolic code.

### 3.2. Necessary and Sufficient Conditions

Given a set $\mathcal{G}_1, \ldots, \mathcal{G}_N$ of N training input graphs where each graph $\mathcal{G} = (V, X, E)$ is an undirected graph with $|V|$ nodes and $|E|$ edges. Each graph $\mathcal{G}_i$ has a property $y_i \in \mathbb{R}$. We follow the similar terminology as (Zhao et al., 2025). The full joint generative model for graph generation $G_{0:T}$ with masking or replacing probability $M_{0:T}$ is as follows:

$$p_\theta(G_{0:T}, M_{0:T}) = p(G_T, M_T) \prod_{t=1}^{T} p_\theta(G_{t-1}, M_{t-1} \mid G_t, M_t)$$

(1)

We use the language of necessary and sufficient conditions in mathematics to justify our methodology. Conventional methods for diffusion are sufficient that use only neural learning. Thus, equation (1) is just a neural formulation and we have ignored symbolic representation which is at the data preprocessing level. The neural formulation collapses all varieties of atom types to a one hot vector at the input of the model. Thus, the denoiser in the neural model is not enough expressive for denoising, and message passing

networks like GNN or even the graph transformer assume the input data is already in a good expressive representation while a one hot representation does not capture contextual information around each node or edge. The neuro-symbolic model can distinguish different types of a particular atom type or edge type even before feeding it into the neural model and the design begins with data-level considerations. The neuro-symbolic joint model could be written as:

$$p_\theta(G_{0:T}, M_{0:T}, S) = p_\theta(G_{0:T}, M_{0:T} \mid S)p_\phi(S)$$

(2)

The symbolic model $S$ in (2) has parameter $\phi$. We have modeled the symbolic model as a tokenizer using VQ-VAE framework. The output of symbolic model is the learned tokens which is fed into the neural model.

### 3.3. Modeling Outline

We first design a neural diffusion model called SAD (structure aware diffusion) that learns how to use structural information of the molecule to learn the scheduling of the forward process. Note that the denoiser of SAD is purely neural as is shown in figure 3 (a). We also design a neurosymbolic model called VQ-SAD which is exactly similar to SAD except having two differences. The first difference is that the replacing probability $\gamma_t$ is also considered. The second difference is that it has a symbolic model to tokenize both atom types and bond types. Once the symbolic model is trained, it is frozen, and the learned codes are fed to the neural model. The neuro-symbolic training view of VQ-SAD is shown in figure 3 (c). We got unstable training for the case of figure 3 (b) that trains both symbolic and neural model simultaneously.

Instead of just relying on hand crafted features like cycle count or degree in (Vignac et al., 2022), we first provide a systematic approach for modeling structural features of each node and edge using relative random walk probabilities (RRWP). Then the structural embedding is affecting the node and edge schedulars as well as the the GNN denoiser to predict the node and edge types. After the model is trained, the sampling algorithm generates graphs either unconditionally or based on the given target properties. The framework of SAD is shown in figure 4

### 3.4. Structural Embedding

To model structural information efficiently, SAD uses RRWP which is introduced in (Ma et al., 2023), (Dwivedi et al., 2022), (Rampášek et al., 2022). Let $A \in \mathbb{R}^{n \times n}$ denote the adjacency matrix of a graph $(V, E)$ with $n$ nodes, and let $D$ be the diagonal degree matrix. We define

$$M := D^{-1}A,$$

(3)

such that each element $M_{ij}$ represents the probability of transitioning from node $i$ to node $j$ in a single step of a

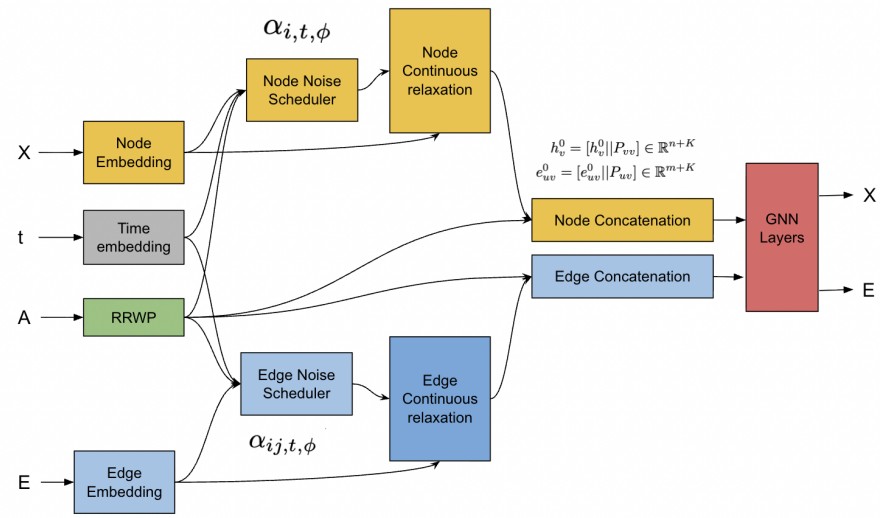

*Figure 4.* The framework of SAD.

simple random walk. The *Relative Random Walk Probabilities (RRWP)* positional encoding is defined for every pair of nodes $i, j \in V$ as:

$$P_{i,j} = [I, M, M^2, \ldots, M^{K-1}]_{i,j} \in \mathbb{R}^K, \quad (4)$$

where $I$ denotes the identity matrix. For each node $i \in V$, the diagonal term $P_{i,i}$ can additionally be used as an initial node-level structural encoding.

### 3.5. Diffusion

#### 3.5.1. CATEGORICAL NOISE

We assume each discrete random variable $x_t$ has a categorical distribution $x_t \sim \text{Cat}(x_t; p)$ with $p \in [0, 1]^K$ and $1^T p = 1$. The forward process with discrete variables $q(x_t|x_{t-1})$ can be represented as a transition matrix $Q_t \in [0, 1]^{K \times K}$ such that $[Q_t]_{ij} = q(x_t = e_j | x_{t-1} = e_i)$. Then we have

$$q(x_t|x_{t-1}) = \text{Cat}(x_t; Q_t^T x_{t-1}) \quad (5)$$

Given transition matrices $Q_1, \ldots, Q_T$, the forward conditional marginal distribution is traditionally modeled as:

$$\begin{aligned} q(x_t|x_0) &= \text{Cat}(x_t; \bar{Q}^T x_0) \\ \bar{Q}_t &= Q_1 \ldots Q_t \\ Q_t &= \alpha_t I + (1 - \alpha_t) \mathbf{1} m^T \end{aligned} \quad (6)$$

where $\alpha_t \in [0, 1]$, and $m$ could be one-hot vector for the mask token or a uniform distribution ($\frac{1}{K_v}$ for nodes and $\frac{1}{K_e}$ for edges) for the case of resampling from uniform distribution. SAD uses the one hot vector for the mask token. These static forward distribution neglects different states

of the noising process. To this end, we define learnable Markov kernels for node types as follows:

$$q_\phi(x_t^i \mid x_{t-1}^i) = \begin{cases} \beta_{t,\phi}^i, & \text{if } x_t^i = [\text{mask}] \\ 1 - \beta_{t,\phi}^i, & \text{if } x_t^i = x_{t-1}^i \end{cases} \quad (7)$$

Similarly, we can define for edge transitions:

$$q_{\phi'}(e^{ij,t} \mid e^{ij,t-1}) = \begin{cases} \beta_{t,\phi'}^{'ij}, & \text{if } e_t^{ij} = [\text{mask}] \\ 1 - \beta_{t,\phi'}^{'ij}, & \text{if } e_t^{ij} = e_{t-1}^{ij}. \end{cases} \quad (8)$$

where $\beta$ and $\beta'$ are node and edge schedular respectively. In diffusion modeling, the parameters $\alpha_t$ and $\beta_t$ are related through the equation

$$\alpha_t = 1 - \beta_t, \quad (9)$$

where $\beta_t$ denotes the noise variance at step $t$.

#### 3.5.2. ADAPTIVE SCHEDULING

**Learnable structure-aware embeddings.** Since the embeddings in SAD are not permutation invariant and depend on the node ordering, it is essential to incorporate structural information of the graph into the noise scheduling process. Without such information, symmetric nodes or edges may receive identical noise schedules, leading to state clashing during diffusion. A straightforward approach is to utilize graph positional encodings; however, these often fail to distinguish motifs, such as cyclic structure. To address this issue, we concatenated RRWP to the embeddings to calculate the scheduling as follows:

$$\text{node schedular output} := f\left(W^{(n)}\mathbf{h}_v, W_n^{(s)}\mathbf{P}_{vv}, \mathbf{c}\right) \in \mathbb{R}^K$$

$$\text{edge schedular output} := f'\left(W^{(e)}\mathbf{e}_{uv}, W_e^{(s)}\mathbf{P}_{uv}, \mathbf{c}\right) \in \mathbb{R}^K \quad (10)$$

where $f(\cdot)$ is a learnable function with $K$ outputs that integrates both the node-wise embeddings and their structural context. Our noise embedding network $f(\cdot)$ computes the corresponding noise scheduling. $\mathbf{c}$ is added in equation (10) for the case of conditional generation as a property representation. This design ensures that the scheduling network receives step-specific, structure-aware representations of graph elements, thereby reducing the likelihood of state clashing. We introduce a new method for scheduling called adaptive scheduling which adapts the noise scheduling based on the time encoding, node and edge type as well as the structural encoding of the node and edges. $W^{(n)}$, $W^{(e)}$, $W^{(s)}$ in equation 10 are learnable node, edge, and structural embeddings respectively before feeding them to the learnable network $f(\cdot)$. We denote the $k$-th output of network for the $i$-th node as $f_k^i$ and define the node schedule:

$$\alpha_t^i = \sigma(-\zeta_t^i) \tag{11}$$

where $\sigma(.)$ is the sigmoid function, and $\zeta_t^i$ is the noise scheduling network which is computed as follows:

$$\zeta_t^i = \hat{\zeta}_t^i \cdot (\zeta_{max} - \zeta_{min}) + \zeta_{min}$$
$$\hat{\zeta}_t^i = \frac{\sum_{k=1}^{K} f_k^i t^k}{\sum_{k=1}^{K} f_k^i} \tag{12}$$

Note that edge schedules $\alpha'$ can be defined similar to node schedules $\alpha$ by using the outputs of $f'(\cdot)$.

### 3.6. Denoising

We use equation (4) to concatenate node and edge features with their corresponding structural features.

$$h_v^0 = [h_v^0 || P_{vv}] \in \mathbb{R}^{n+K}$$
$$e_{uv}^0 = [e_{uv}^0 || P_{uv}] \in \mathbb{R}^{m+K} \tag{13}$$

For denoising, we use the method of (Hu et al., 2020), a variant of the Graph Isomorphism Network that also incorporates edge features. The implementation details are provided in the appendix A. The node–type prediction probability is defined as

$$p_\theta(x_v \mid g_t) = \frac{\exp(\hat{y}_v[x_v])}{\sum_{c=1}^{C_n+1} \exp(\hat{y}_v[c])}, \tag{14}$$

where $x_v$ is the ground truth node label and $C_n$ is the number of node types. The logits $\hat{y}_v \in \mathbb{R}^{C_n+1}$ are obtained from the node embedding $h_v \in \mathbb{R}^d$ via a two layer MLP:

$$\hat{y}_v = W^{(2)} \sigma\left(W^{(1)} h_v + b^{(1)}\right) + b^{(2)}, \tag{15}$$

where $W^{(1)}, W^{(2)}$ are weight matrices, $b^{(1)}, b^{(2)}$ are bias vectors, and $\sigma$ denotes the ReLU activation function. Similarly, the edge type prediction probability is

$$p_\theta(y_{uv} \mid g_t) = \frac{\exp(\hat{y}_{uv}[y_{uv}])}{\sum_{c=1}^{C_e+1} \exp(\hat{y}_{uv}[c])}, \tag{16}$$

where $y_{uv}$ is the true edge-type label and $C_e$ is the number of edge types. The edge logits $\hat{y}_{uv} \in \mathbb{R}^{C_e+1}$ are computed from the concatenation of node and edge embeddings:

$$\hat{y}_{uv} = W^{(4)} \sigma\left(W^{(3)} [h_u || h_v || h_{uv}] + b^{(3)}\right) + b^{(4)}, \tag{17}$$

where $h_{uv}$ is the edge embedding.

---

**Algorithm 1** Training Algorithm for SAD

---

**Require:** Graph dataset $\mathcal{G}$, node types $C_n$, edge types $C_e$, learning rate $\eta$, number of epochs $E$
**Require:** Node and edge embedding networks $f_\theta$, noise schedule parameters $\alpha_{t,\phi}$
**Ensure:** Trained model parameters $\theta, \phi$
1: **for** epoch = 1 **to** $E$ **do**
2:     **for** each graph $g$ in dataset $\mathcal{G}$ **do**
3:         Sample a diffusion timestep $t$ from uniform distribution
4:         Compute structural embedding for nodes and edges using equation (4), (13)
5:         Apply forward diffusion to graph $g$ to get noisy graph $g_t$
6:         Compute node predictions $\hat{y}_v = f_\theta^{node}(g_t)$
7:         Compute edge predictions $\hat{y}_{uv} = f_\theta^{edge}(g_t)$
8:         Compute the loss $\mathcal{L}$ in equation (18)
9:         Update parameters: $\theta \leftarrow \theta - \eta \nabla_\theta \mathcal{L}(\theta, \phi)$
10:       Update noise schedule parameters: $\phi \leftarrow \phi - \eta \nabla_\phi \mathcal{L}(\theta, \phi)$
11:     **end for**
12: **end for**

---

### 3.7. Training of SAD

Inspired by continuous extension of simplified negative ELBO (NELBO) in (Sahoo et al., 2024), (Shi et al., 2024), we define the following noise conditioned loss:

$$\mathcal{L}(\theta, \phi) := -\mathbb{E}_{t,g,g_t} \left[ \sum_{i=1}^{N} \frac{\dot{\alpha}_{i,t,\phi}}{1 - \alpha_{i,t,\phi}} \log p_\theta(x_i \mid g_t) \right.$$
$$\left. + \lambda \sum_{i,j=1}^{N} \frac{\dot{\alpha}_{ij,t,\phi}}{1 - \alpha_{ij,t,\phi}} \log p_\theta(e_{ij} \mid g_t) \right] \tag{18}$$

The noise schedule $\alpha_t$ can be defined either as a discrete product over the timesteps or as a continuous function of time, i.e., a continuous schedule $\alpha(t)$:

$$\alpha_{i,t,\phi} = \prod_{s=1}^{i,t,\phi} (1 - \beta_s^i)$$
$$\alpha_{ij,t,\phi} = \prod_{s=1}^{ij,t,\phi} (1 - \beta_s'^{ij}) \tag{19}$$

Note that $\dot{\alpha}$ in equation (18) is calculated by linear interpolation followed by automatic differentiation in Pytorch. Training with recursive sampling is computationally expensive. Thus, in practice, we directly predict the original graph $g_0$ from the noisy intermediate state $g_t$ by sampling a random time $t$. The full algorithm for training is shown in algorithm 1.

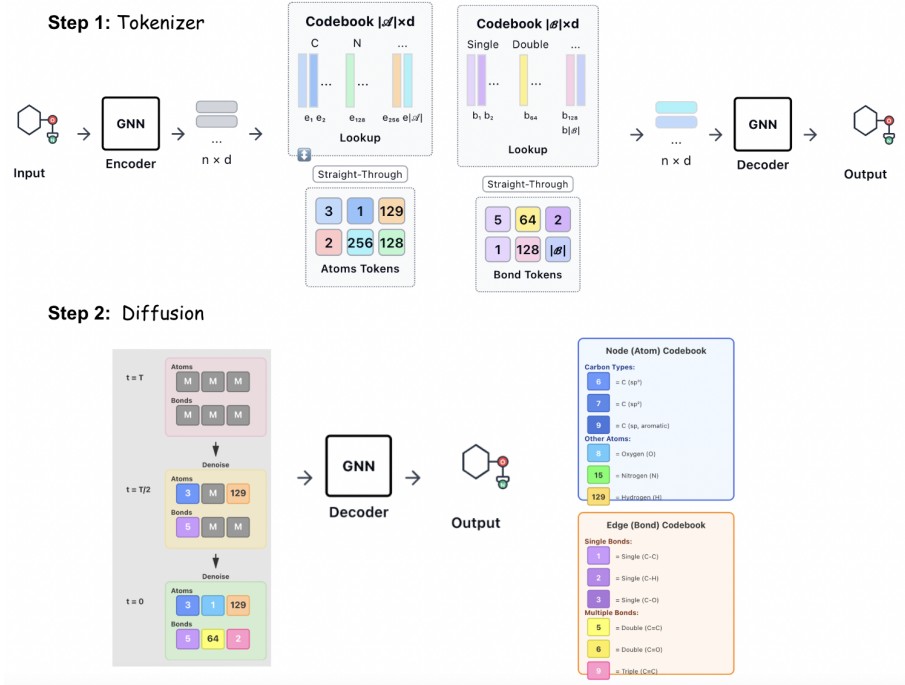

*Figure 5.* The steps of VQ-SAD.

### 3.8. VQ-SAD

VQ-SAD is built on top of SAD with two major differences. The first one is generalizing the learnable transition matrix to consider the case of having replacement probability $\gamma$. The second major difference is using VQ-VAE as a tokenizer to create a more balanced atom and edge types. The training of VQ-SAD is shown in algorithm 2 and the learning of tokenizer details are in appendix C. The architecture of VQ-SAD is shown in figure 5. The algorithm consists of two steps. In the first step, a VQ-VAE model is trained to learn the codes in the latent space for both node types and edge types. The second step adopts the tokenizer using the same learned encoder in the first step, and then denoises the embeddings till the graph has no masked node and masked edges. The mask-and-replace diffusion transition matrix at step $t$ is:

$$
Q_t = \begin{bmatrix}
\alpha_{t,\psi} & \gamma_{t,\phi} & \gamma_{t,\phi} & \cdots & \beta_t \\
\gamma_{t,\phi} & \alpha_{t,\psi} & \gamma_{t,\phi} & \cdots & \beta_t \\
\gamma_{t,\phi} & \gamma_{t,\phi} & \alpha_{t,\psi} & \cdots & \beta_t \\
\vdots & \vdots & \vdots & \ddots & \vdots \\
0 & 0 & 0 & \cdots & 1
\end{bmatrix} \quad (20)
$$

Let $e_i$ denote the one-hot vector corresponding to state $i$, i.e., a vector with $1$ at position $i$ and $0$ elsewhere. Then, the cumulative transition matrix up to step $t$ applied to an initial vector $v(x_0)$ is:

$$
\overline{Q}_t v(x_0) = \overline{\alpha}_t v(x_0) + \overline{\gamma}_t \frac{1}{K-1} \sum_{i \neq x_0} e_i + \overline{\beta}_t v(K+1) \quad (21)
$$

where the cumulative coefficients are:

$$
\begin{aligned}
\overline{\alpha}_t &= \prod_{i=1}^{t} \alpha_i, \\
\overline{\beta}_t &= 1 - \prod_{i=1}^{t}(1-\beta_i), \\
\overline{\gamma}_t &= 1 - \overline{\alpha}_t - \overline{\beta}_t.
\end{aligned} \quad (22)
$$

Note that $\gamma$ in (22) is the replacement probability which is essential to correct the node and edge types that are unmasked incorrectly. Algorithm 3 in appendix B shows the molecule generation of VQ-SAD. It requires pretrained encoder of VQ-VAE to map from original node types to the code space. It also receives the pretrained decoder of VQ-VAE to map from code space back to the original node and edge types.

**Algorithm 2** The pseudo code for training VQ-SAD

1: **Input:** Molecular dataset $\{G_1, \ldots, G_N\}$ with nodes $V$ and edges $E$
2: **Step 1: Train VQ-VAE Tokenizer**:
3: Run Algorithm 6
4: Freeze the weights and save this model as a pretrained model
5: Load Pretrained Model: Atom codebook $E_{\text{atom}}$, bond codebook $E_{\text{bond}}$
6: **Step 2: Train Discrete Diffusion in Latent Space**
7: Convert molecules to discrete codes $(Z_V, Z_E)$ using pretrained encoders and codebooks
8: Initialize denoising model $D$
9: $H_V = f_{\text{enc}}^{\text{node}}(V)$ {encode nodes}
10: $H_E = f_{\text{enc}}^{\text{edge}}(E)$ {encode edges}
11: $Z_V = \text{Quantize}(H_V, E_{\text{atom}})$ {atom codes}
12: $Z_E = \text{Quantize}(H_E, E_{\text{bond}})$ {bond codes}
13: **for** each diffusion step $t$ and molecule codes $(Z_V, Z_E)$ **do**
14: $\quad (Z_V^{\text{noisy}}, Z_E^{\text{noisy}}) = \text{Diffuse}(Z_V, Z_E, t)$
15: $\quad (Z_V^{\text{pred}}, Z_E^{\text{pred}}) = D(Z_V^{\text{noisy}}, Z_E^{\text{noisy}}, t)$
16: $\quad$ Compute node predictions $\hat{y}_v = f_\theta^{node}(Z_V)$
17: $\quad$ Compute edge predictions $\hat{y}_{uv} = f_\theta^{edge}(Z_E)$
18: $\quad$ Compute the loss $\mathcal{L}$ in equation (18)
19: $\quad$ Update parameters: $\theta \leftarrow \theta - \eta \nabla_\theta \mathcal{L}_{\text{total}}(\theta, \phi)$
20: $\quad$ Update noise schedule parameters: $\phi \leftarrow \phi - \eta \nabla_\phi \mathcal{L}_{\text{total}}(\theta, \phi)$
21: **end for**
22: **Output:** denoising model $D$

---

**Algorithm 3** Graph Generation in VQ-SAD (Sampling Procedure)

**Require:** Number of steps $T$, trajectory network $\Gamma_\phi$, denoising model $p_\theta$, encoder and decoder of VQ-VAE
1: Initialize graph $x_T \leftarrow$ fully masked graph
2: Compute structural embedding of each node and edge using RRWP
3: Map original node and edge types using encoder of pretrained VQ-VAE
4: **for** $t = T, T-1, \ldots, 1$ **do**
5: $\quad$ Compute masking schedule for $\alpha$, $\beta$, and $\gamma$
6: $\quad$ Predict categorical distributions using the denoising model $p_\theta$:
$$p_\theta(v), \quad p_\theta(e_{uv})$$
7: $\quad$ Update $x_{t-1}$ based on $\alpha$, $\beta$, and $\gamma$
8: **end for**
9: Compute the latent vector of each node and edge from embedding network
10: Compute the graph using decoder of learned VQ-VAE
11: **return** Complete graph $x_0$

---

*Table 1.* Generation quality on QM9 with explicit hydrogens.

| Model | Valid. ↑ | Uni. ↑ | FCD ↓ | NSPDK ↓ |
|---|---|---|---|---|
| DIGRESS | 95.35 | 96.45 | 0.35 | 0.001 |
| MELD | 95.73 | 96.82 | 0.34 | 0.0008 |
| SAD | 96.52 | 97.16 | 0.35 | 0.0008 |
| VQ-SAD | **97.31** | **98.51** | **0.31** | **0.0007** |

*Table 2.* Generation quality on ZINC250k.

| Model | Validity ↑ | Uni. ↑ | FCD ↓ | NSPDK ↓ |
|---|---|---|---|---|
| DIGRESS | 91.03 | 92.62 | 1.37 | 0.012 |
| MELD | 92.83 | 93.19 | 1.27 | 0.011 |
| SAD | 93.01 | 94.28 | 1.38 | 0.011 |
| VQ-SAD | **93.84** | **94.73** | **1.21** | **0.010** |

## 4. Experiments

### 4.1. Dataset

We evaluate our molecular generative model on two widely used benchmarks, QM9 and ZINC250k. QM9 (Ramakrishnan et al., 2014) is a curated subset of the GDB-17 chemical universe consisting of approximately 134,000 stable small organic molecules with up to nine heavy atoms (C, N, O, F), provided together with a rich set of quantum-chemical properties computed at the DFT level. ZINC250k (Gómez-Bombarelli et al., 2018) is a subset of roughly 250,000 drug-like molecules sampled from the larger ZINC database, filtered by physicochemical constraints such as logP and synthetic accessibility. Together, these datasets provide complementary evaluation regimes. QM9 emphasizes coverage and fidelity on a constrained space of small organic molecules, whereas ZINC250k probes scalability to larger, more structurally complex, and pharmaceutically meaningful compounds. See appendix E for more comparison.

### 4.2. Metrics

In accordance with the evaluation protocols established in prior studies (Jo et al., 2024), (Jo et al., 2022), (Liu et al., 2024), (Seo et al., 2025), we assess the effectiveness of our framework through two categories of metrics.

*Table 3.* Performance comparison of different methods on validity and uniqueness metrics for Heat Capacity at Constant Volume $C_v$ and Dipole Moment $\mu$ for QM9 dataset

| | Validity ↑ | Uni. ↑ | Validity ↑ | Uni. ↑ |
|---|---|---|---|---|
| | $C_v$ | | $\mu$ | |
| **DiGress** | 91.64 | 88.25 | 90.12 | 85.93 |
| **MELD** | 93.53 | 88.25 | 90.42 | 86.85 |
| **SAD** | 94.82 | 91.57 | 80.83 | 91.88 |
| **VQ-SAD** | 95.21 | 92.64 | 91.75 | 91.22 |

**Unconditional Generation.** For unconditional molecule generation, we produce 1000 samples and measure their performance across four standard criteria:

1. **Validity (Valid.)**: the percentage of molecules that are chemically valid.

*Table 4.* **Collision Rate Comparison** between VQ-SAD and MELD models on molecular datasets. Lower is better ($\downarrow$).

| Dataset | Model | Collision Rate $\downarrow$ |
|---------|-------|-----------------------------|
| QM9 | MELD | 0.35 |
| | VQ-SAD | 0.21 |
| ZINC250k | MELD | 0.27 |
| | VQ-SAD | 0.18 |

2. **Uniqueness**: the proportion of valid molecules that are structurally distinct within the generated set.

3. **Fréchet ChemNet Distance (FCD)** (Preuer et al., 2018): a distribution-level similarity metric based on ChemNet embeddings between generated and reference sets.

4. **NSPDK** (Costa & De Grave, 2010): a graph kernel score that evaluates the topological resemblance to the reference molecules.

Table 1 illustrates the baseline comparison for unconditional generation of QM9 dataset. Table 2 shows the results for unconditional generation on ZINC250k. Figure 6 shows the generated molecules from QM9 dataset without using structural information. The skipped molecule indices show the molecules that are invalid and are skipped. Using structural information provides more valid molecules as are shows in figure 7, 8. Similarly, figure 9 shows the generated molecules from ZINC250k dataset without using structural information. Using structural information provides more valid molecules as are shows in figure 10, 11.

**Conditional Generation.** For conditional generation, we generate 1000 samples and evaluate them for two properties in QM9 dataset. We followed (Ho & Salimans, 2021) for a classifier free conditional generation but using GNN for the denoiser model. Table 3 shows the performance comparison of baselines for Heat Capacity at Constant Volume $C_v$ and Dipole Moment $\mu$ for QM9 dataset. One limitation of our work for conditional generation is that generated graph samples become sharper and more aligned with the desired property prompt, but the model collapses onto a narrow set of high-likelihood modes and reduces graph diversity.

### 4.3. Ablation Study

MELD identified the critical cause of an issue known as a state-clashing problem where the forward diffusion of distinct molecules collapse into a common state, which results a mixture of reconstruction targets that cannot be learned using typical reverse diffusion process. If all node embeddings collapse, model produces graphs with identical node types and repeated graph patterns which limits the

---

**Algorithm 4** Node Collision Rate

**Require:** Embeddings $\{H^{(t)}\}_{t=1}^{T}$, threshold $\varepsilon$
**Ensure:** Collision rate $CR$
1: $C \leftarrow 0$
2: $N \leftarrow 0$
3: **for** $t = T, T-1, \ldots, 1$ **do**
4:    **for** $i = 1$ to $n$ **do**
5:       **for** $j = i+1$ to $n$ **do**
6:          $N \leftarrow N+1$
7:          **if** $\|h_i^{(t)} - h_j^{(t)}\|_2 < \varepsilon$ **then**
8:             $C \leftarrow C+1$
9:          **end if**
10:       **end for**
11:    **end for**
12: **end for**
13: $CR \leftarrow C/N$

---

model expressiveness. With this motivation, we studied the effect of using RRWP rather than just using degree information in the denoising process. We can infer from our performance measures that VQ-SAD can reduce the state-clashing problem by providing informative structural information that can guide the diffusion. To systematically analyze this problem we define the following collision rate to measure the fraction below a fixed threshold. Let $H^{(t)} = \{h_1^{(t)}, \ldots, h_n^{(t)}\}$ denote the continuous node embeddings at reverse diffusion step $t$, where $h_i^{(t)} \in \mathbb{R}^d$. A collision occurs when

$$\|h_i^{(t)} - h_j^{(t)}\|_2 < \varepsilon.$$

The full algorithm for calculating collision rate is shown in algorithm 4. It can be observed from table 4 that collision rate of VQ-SAD is slightly lower than MELD since the tokenizer of VQ-SAD avoids collapsing contextually different nodes with the same atom types to collapse into the same one hot representation at the input and therefore similar node representations at different steps of diffusion.

## 5. Conclusion

We identified that previous molecular generation methods were only focused on neural modeling and ignored symbolic information in the chemistry of molecules. VQ-SAD is a neuro-symbolic method that models the symbolic information as a tokenizer for each atom and edge types. The neural model then uses these symbolic features to remove the problem of unbalanced categories in node and edge types of QM9 and ZINC250k datasets. Unlike most diffusion models, VQ-SAD learns the forward process using staying probability, replacement probability as well as masking probability. The differentiability of forward process is achieved by using gumbel softmax. The scheduling networks for VQ-SAD are learned using structural modeling based on RRWP.

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

## A. Denoising Implementation

To perform denoising, we adopt the edge-enhanced Graph Isomorphism Network (GIN) variant proposed by (Hu et al., 2020). This model extends the original GIN by incorporating edge features into the neighborhood aggregation process, enabling more expressive message passing over molecular graphs. Below, we provide the formulation of the update rule together with detailed definitions of all components.

$$h_v^{(k)} = \text{MLP}^{(k)}\Big( \Big(1 + \epsilon^{(k)}\Big) \cdot h_v^{(k-1)} + \sum_{u \in \mathcal{N}(v)} \text{ReLU}\left(h_u^{(k-1)} + e_{uv}\right)\Big). \quad (23)$$

$h_v^{(k)}$ is the Node representation of $v$ at the $k$-th layer. $h_v^{(k-1)}$ is the node representation of $v$ from the $(k-1)$-th layer. $\epsilon^{(k)}$ is the learnable parameter controlling the relative contribution of self-features and aggregated neighbor features. $\mathcal{N}(v)$ is Set of neighboring nodes of $v$. $e_{uv}$ is Feature vector associated with the edge between nodes $u$ and $v$. $\text{ReLU}(\cdot)$ is the rectified linear unit activation function. Finally, $\text{MLP}^{(k)}(\cdot)$ is the multilayer perceptron at the $k$-th layer for feature transformation.

## B. Graph Generation

The graph generation of SAD is shown in algorithm 5.

---
**Algorithm 5** Graph Generation in SAD (Sampling Procedure)

---
**Require:** Number of steps $T$, trajectory network $\Gamma_\phi$, denoising model $p_\theta$
1: Initialize graph $x_T \leftarrow$ fully masked graph
2: **for** $t = T, T-1, \ldots, 1$ **do**
3:    Compute structural embedding of each node and edge using RRWP
4:    Compute masking schedule $\gamma_t \leftarrow \Gamma_\phi(t)$
5:    Identify subset of nodes/edges to unmask according to $\gamma_t$
6:    Predict categorical distributions using the denoising model $p_\theta$ :
$$p_\theta(v), \quad p_\theta(e_{uv})$$
7:    Sample node/edge types for masked positions from $p_\theta$
8:    Update $x_{t-1}$ by replacing corresponding masks with sampled values
9: **end for**
10: **return** Complete graph $x_0$

---

## C. VQ-VAE Tokenizer Loss in VQ-SAD

Let $G$ be a molecule graph with atoms $V = \{v_1, \ldots, v_n\}$. For each atom $v_i$, the encoder outputs

$$h_i = f_{\text{enc}}(v_i). \tag{24}$$

The codebook is $\{e_k\}_{k=1}^K \subset \mathbb{R}^D$, and the quantized embedding is as follows:

$$z_i = \arg\min_k \|h_i - e_k\|_2^2, \qquad e_{z_i} \text{ is selected.} \tag{25}$$

The decoder reconstructs :

$$\hat{v}_i = f_{\text{dec}}(e_{z_i}). \tag{26}$$

The VQ-VAE loss with the scaled cosine term is as follows:

$$\mathcal{L}_{\text{VQ}} = \frac{1}{n} \sum_{i=1}^n \left[ \left( 1 - \frac{v_i^\top \hat{v}_i}{\|v_i\| \, \|\hat{v}_i\|} \right)^\gamma \right.$$
$$+ \left\| \text{sg}(h_i) - e_{z_i} \right\|_2^2 \tag{27}$$
$$\left. + \beta \left\| h_i - \text{sg}(e_{z_i}) \right\|_2^2 \right]$$

where $\gamma \geq 1$ is the scaling exponent for the cosine term, $\beta > 0$ is the commitment weight, and $\text{sg}[\cdot]$ denotes stop-gradient.

Let $G$ be a molecule graph with bonds $E = \{e_1, \ldots, e_m\}$. For each bond $e_j$, the encoder outputs

$$h_j^{\text{bond}} = f_{\text{enc}}(e_j). \tag{28}$$

The bond codebook is $\{b_k\}_{k=1}^{K_b} \subset \mathbb{R}^D$, and the quantized embedding is

$$z_j^{\text{bond}} = \arg\min_k \|h_j^{\text{bond}} - b_k\|_2^2, \qquad b_{z_j^{\text{bond}}} \text{ is selected.} \tag{29}$$

The decoder reconstructs :

$$\hat{e}_j = f_{\text{dec}}(b_{z_j^{\text{bond}}}). \tag{30}$$

The VQ-VAE loss for bond types with the scaled cosine term is as follows :

$$\mathcal{L}_{\text{VQ}}^{\text{bond}} = \frac{1}{m} \sum_{j=1}^m \left[ \left( 1 - \frac{e_j^\top \hat{e}_j}{\|e_j\| \, \|\hat{e}_j\|} \right)^\gamma \right.$$
$$+ \left\| \text{sg}(h_j^{\text{bond}}) - b_{z_j^{\text{bond}}} \right\|_2^2 \tag{31}$$
$$\left. + \beta \left\| h_j^{\text{bond}} - \text{sg}(b_{z_j^{\text{bond}}}) \right\|_2^2 \right]$$

where $\gamma \geq 1$ is the scaling exponent for the cosine term, $\beta > 0$ is the commitment weight, and $\text{sg}[\cdot]$ denotes stop-gradient.

---

**Algorithm 6** The pseudo code for training VQ-VAE as a tokenizer for atom types and edge types

---

1: **Input:** Molecular dataset $\{G_1, \ldots, G_N\}$ with nodes $V$ and edges $E$
2: **Initialize:**
3:     Node encoder $f_{\text{enc}}^{\text{node}}$, node decoder $f_{\text{dec}}^{\text{node}}$, atom codebook $E_{\text{atom}} = \{e_1^{\text{atom}}, \ldots, e_K^{\text{atom}}\}$
4:     Edge encoder $f_{\text{enc}}^{\text{edge}}$, edge decoder $f_{\text{dec}}^{\text{edge}}$, bond codebook $E_{\text{bond}} = \{e_1^{\text{bond}}, \ldots, e_L^{\text{bond}}\}$
5: **for** each epoch **do**
6:     **for** each molecule $G$ **do**
7:         $H_V = f_{\text{enc}}^{\text{node}}(V)$ {encode nodes}
8:         $Z_V = \text{Quantize}(H_V, E_{\text{atom}})$ {atom codes}
9:         $H_E = f_{\text{enc}}^{\text{edge}}(E)$ {encode edges}
10:        $Z_E = \text{Quantize}(H_E, E_{\text{bond}})$ {bond codes}
11:        $V_{\text{rec}} = f_{\text{dec}}^{\text{node}}(Z_V)$
12:        $E_{\text{rec}} = f_{\text{dec}}^{\text{edge}}(Z_E)$
13:        Compute node loss $L_{\text{node}} = \text{ReconLoss}(V, V_{\text{rec}}) + \beta \cdot \text{CommitLoss}(H_V, Z_V)$
14:        Compute edge loss $L_{\text{edge}} = \text{ReconLoss}(E, E_{\text{rec}}) + \beta \cdot \text{CommitLoss}(H_E, Z_E)$
15:        $L_{\text{VQ}} = L_{\text{node}} + L_{\text{edge}}$
16:        Update all encoders, decoders, and codebooks with $\nabla L_{\text{VQ}}$
17:     **end for**
18: **end for**
19: **Output:** Atom codebook $E_{\text{atom}}$, bond codebook $E_{\text{bond}}$

---

*Table 5.* Statistics of molecular graph datasets used in generation benchmarks.

| Dataset | # Molecules | Avg. # Atoms | Atom Types |
|---------|-------------|--------------|------------|
| QM9 | 133K | 18.0 | H, C, N, O, F |
| ZINC250k | 250K | 23.2 | C, N, O, S, Cl |

## D. Generated Molecules

## E. Dataset

Table 5 shows that statistics of QM9 and ZINC250k. Note that ZINC250k has larger and more chemically diverse than QM9. It provides challenging learning signal for generative models because structural variety is high. We observed that ZINC250k tests model capacity to learn diverse molecular geometry and topology. QM9 tests baseline modeling quality but may overestimate generalization ability due to limited diversity.

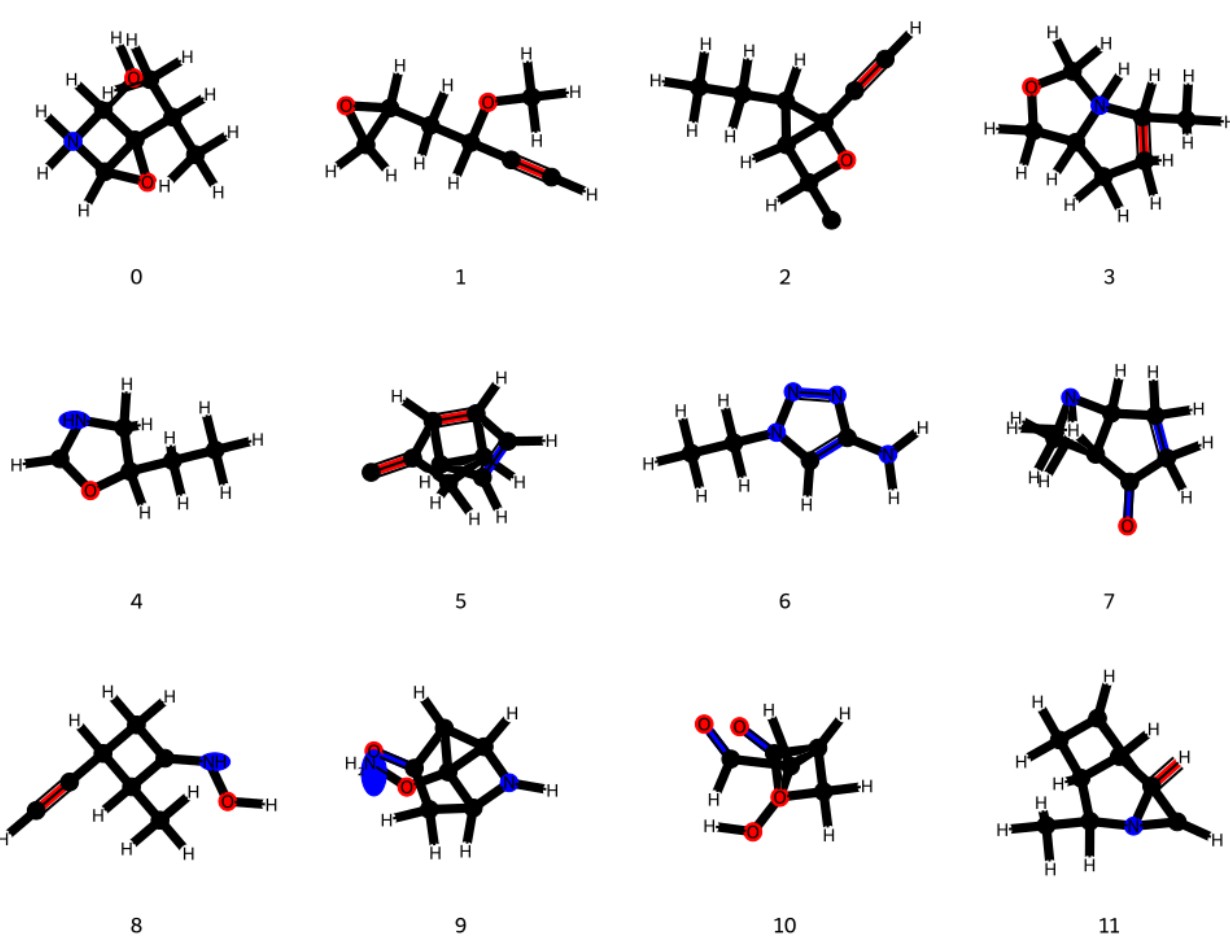

*Figure 6.* generated QM9 without leveraging structural information

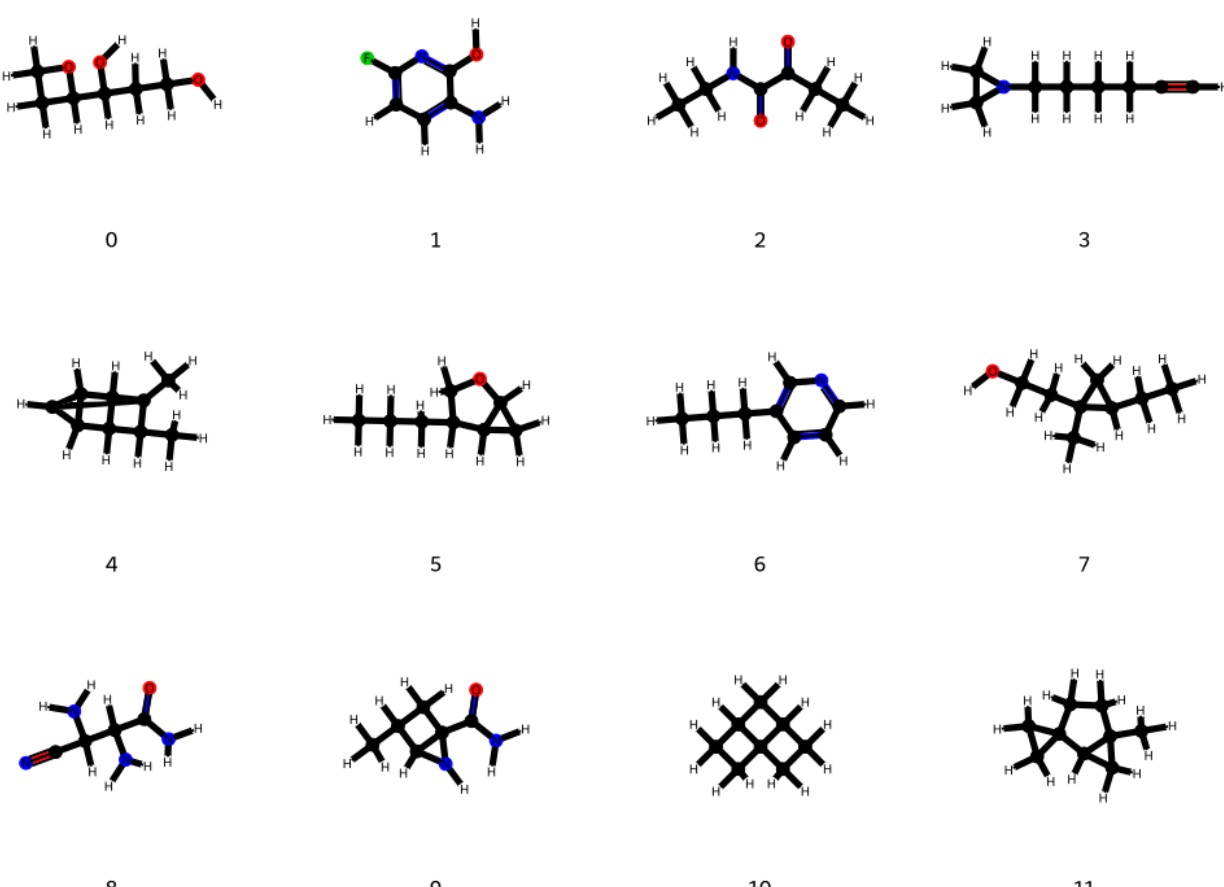

*Figure 7.* generated QM9 using structural information

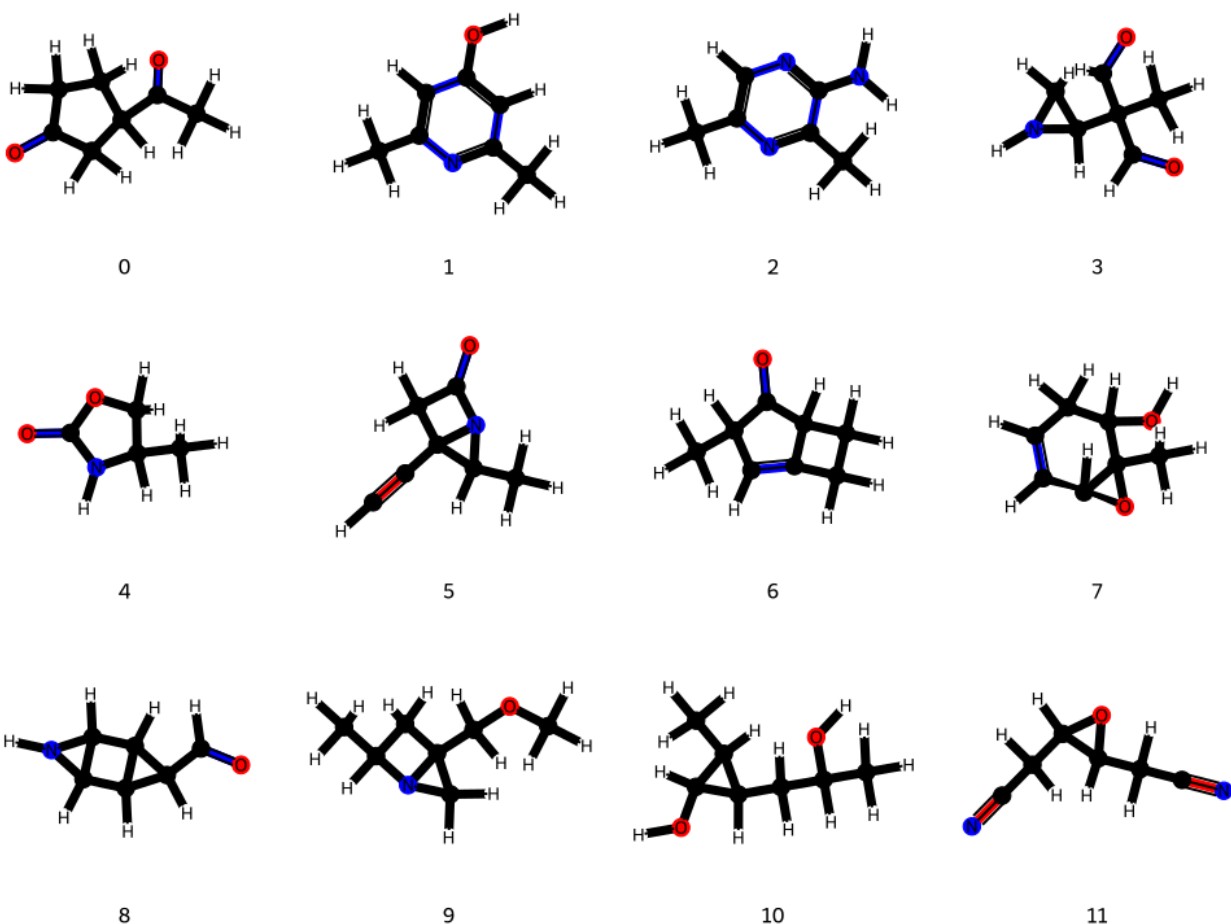

*Figure 8.* generated QM9 using structural information

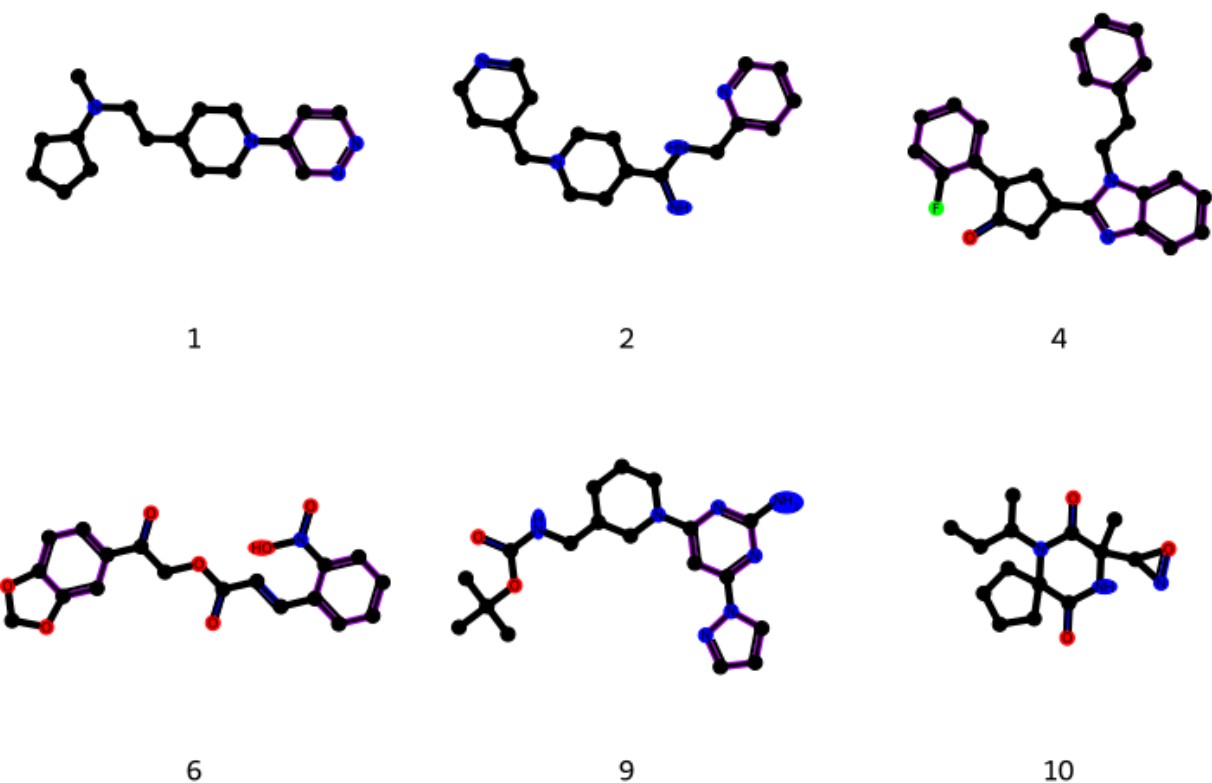

*Figure 9.* generated ZINC250k without leveraging structural information

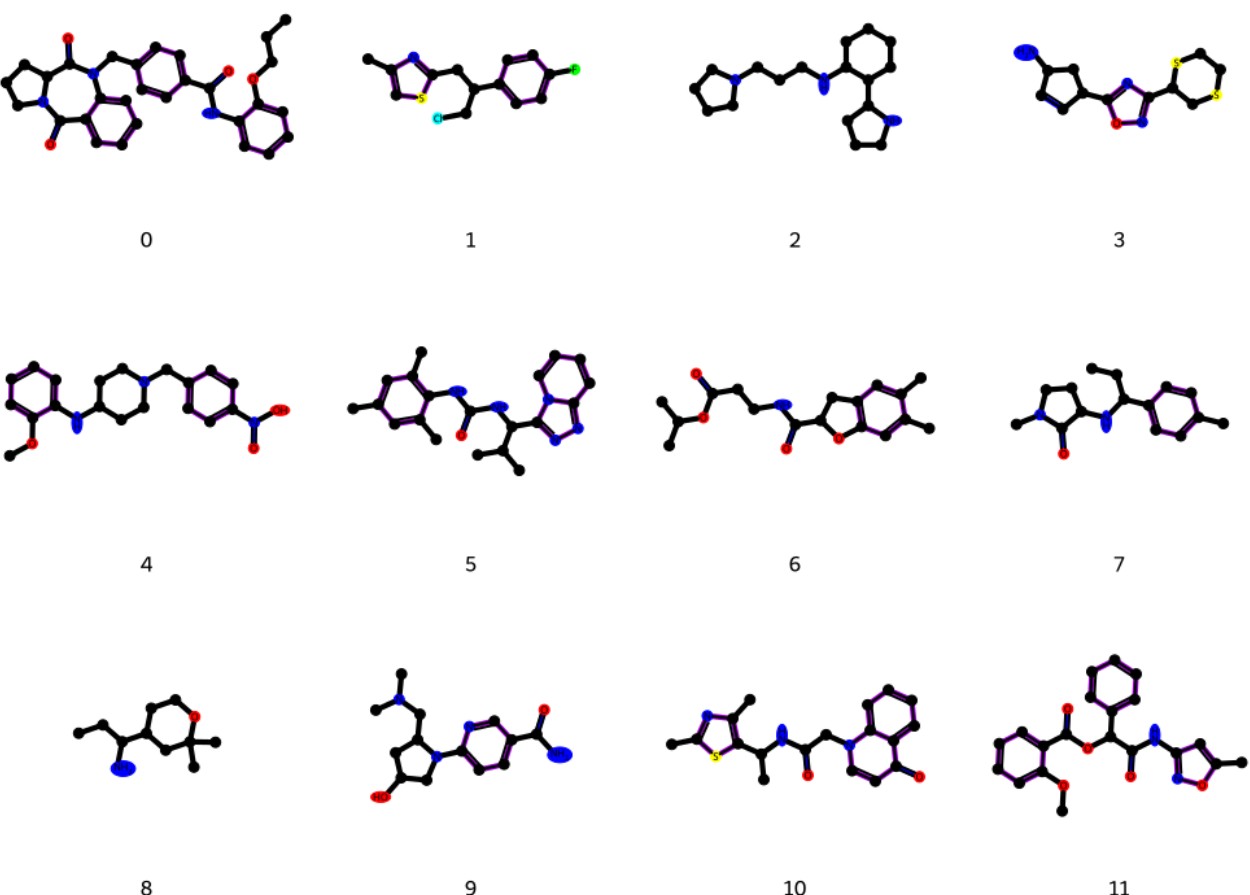

*Figure 10.* generated ZINC250k using structural information

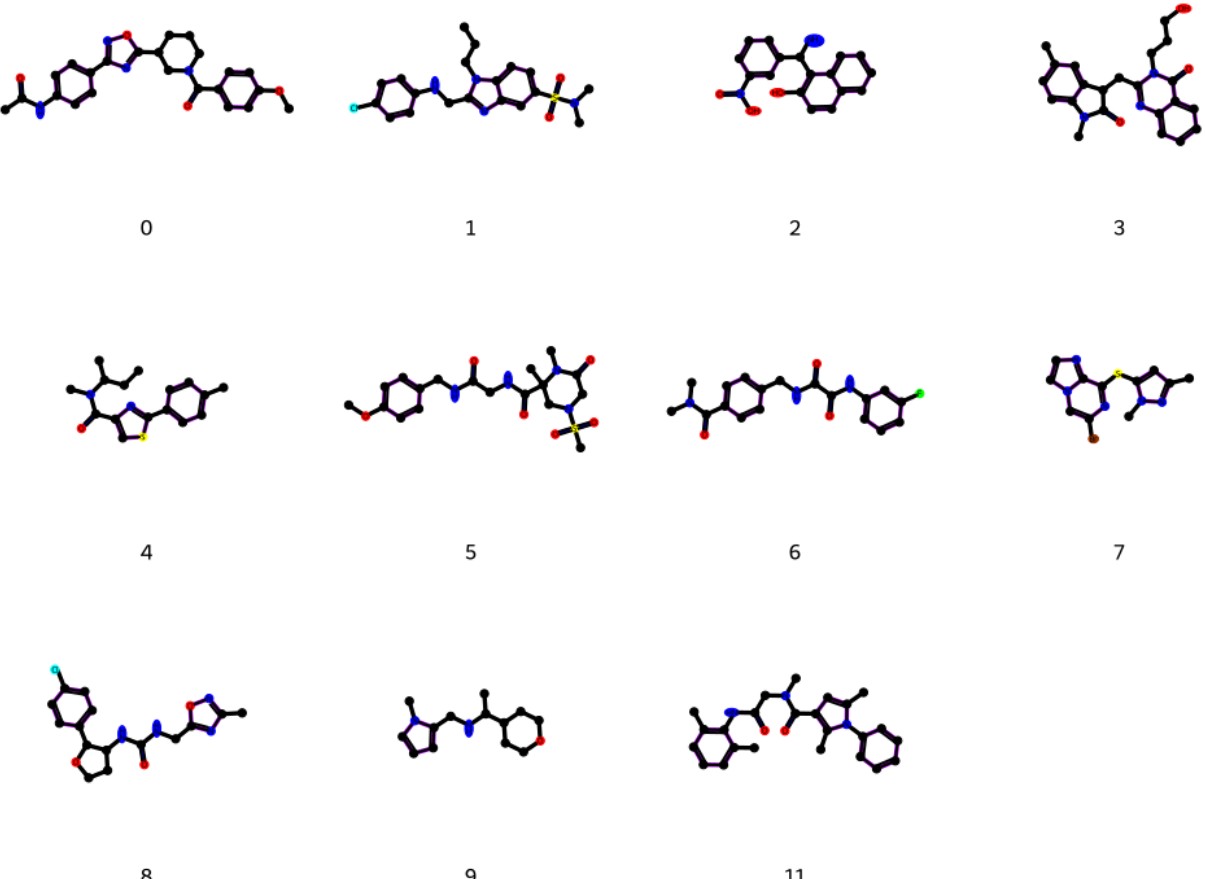

*Figure 11.* generated ZINC250k using structural information

