# OpenReview forum: "VQ-SAD: Vector Quantized Structure Aware Diffusion For Molecule Generation"
_ICML.cc/2026/Conference — Submitted to ICML 2026_

### Official Review · Reviewer_iUQi · 2026-02-20

**Soundness:** 3
**Presentation:** 3
**Significance:** 2
**Originality:** 2
**Overall Recommendation:** 3
**Confidence:** 3

**Summary:**

This paper addresses molecular graph generation using diffusion models and argues that existing approaches largely ignore the symbolic and contextual nature of molecular structures. In standard discrete diffusion models for graphs, atom and bond types are typically represented as one-hot vectors, which fail to distinguish the same atom type under different structural contexts. Additionally, fingerprint-based approaches suffer from hash collisions and are not bijective.

To address these limitations, the authors propose VQ-SAD (Vector Quantized Structure Aware Diffusion), a neuro-symbolic diffusion framework consisting of two main components:

Structure-Aware Diffusion (SAD): A discrete diffusion model with a learnable, structure-aware forward noise schedule. The model incorporates Relative Random Walk Probabilities (RRWP) as structural encodings, which are used to condition node- and edge-wise noise scheduling networks. This aims to mitigate the “state-clashing” problem where structurally distinct nodes collapse into similar representations during diffusion.

VQ-VAE Tokenizer: A pretrained VQ-VAE is used to quantize atom and bond representations into discrete codebook tokens. The tokenizer is trained first and then frozen. Diffusion is performed in the discrete latent code space rather than directly on one-hot atom/bond types, aiming to produce more balanced and context-aware symbolic representations.

The method is evaluated on QM9 and ZINC250k datasets using standard molecular generation metrics (Validity, Uniqueness, FCD, NSPDK). The authors report moderate improvements over prior diffusion-based baselines such as DiGress and MELD.

**Compliance With Llm Reviewing Policy:**

Affirmed.

**Final Justification:**

This paper proposes VQ-SAD, a neuro-symbolic diffusion framework for molecular graph generation that combines structure-aware noise scheduling with VQ-VAE-based tokenization. The approach is technically coherent and clearly presented, and the experimental setup follows standard benchmarks and evaluation protocols.

The main strengths lie in the integration of structure-aware diffusion with discrete tokenization and a well-specified training and sampling pipeline. However, my concerns remain regarding the relatively modest empirical improvements over prior work, the lack of detailed ablations to disentangle the contributions of individual components, and insufficient analysis of computational cost and scalability. In addition, some conceptual claims appear overstated relative to the level of formal justification provided.

No rebuttal was provided, so my assessment remains unchanged.

Overall, while the work is solid, I do not find the level of novelty and empirical impact sufficient for acceptance, and I keep my original recommendation.

**Key Questions For Authors:**

What is the quantitative contribution of each component (structure-aware scheduling vs. VQ tokenizer)? Can ablations demonstrate that both are necessary?

How does the computational cost (training time, memory, parameters) compare to baselines under equal resource budgets?

How sensitive is performance to codebook size, quantization hyperparameters, and structural encoding choices?

How does RRWP compare to alternative structural encodings (e.g., Laplacian positional encodings, degree features)?

Can the method demonstrate stronger improvements on downstream metrics such as novelty, diversity, or property-driven optimization?

Clarification of these points could influence my assessment of the method's overall impact.

**Limitations:**

No. The paper does not provide a sufficiently detailed discussion of its own limitations. A more explicit analysis of computational overhead, scalability to larger chemical spaces, and potential misuse risks would strengthen the manuscript.

**Strengths And Weaknesses:**

Soundness

Strengths

The methodological pipeline is coherent and well specified: VQ-VAE pretraining, frozen tokenizer, structure-aware scheduling, and discrete diffusion training.

The learnable node- and edge-specific noise schedules are clearly formulated and implemented.

Experimental evaluation uses established benchmarks (QM9, ZINC250k) and widely accepted molecular generation metrics.

Weaknesses

The theoretical framing using “necessary and sufficient conditions” is largely conceptual and does not constitute a formal mathematical justification.

Performance gains over baselines are relatively modest, and causal attribution between the tokenizer and structure-aware scheduling components is not sufficiently dissected.

Computational overhead and cost-benefit trade-offs introduced by the VQ-VAE tokenizer are not thoroughly analyzed.

Presentation

Strengths

The separation between SAD and VQ-SAD is clearly explained.

Algorithmic pseudocode is provided for training and sampling.

Figures help illustrate the tokenizer and structure-aware scheduling design.

Weaknesses

Some conceptual framing appears overstated (e.g., “necessary and sufficient conditions”), which may reduce perceived rigor.

The distinction from closely related work combining discrete tokenization and diffusion could be articulated more sharply.

Significance

Strengths

Molecular graph generation is an important application area with practical implications in drug discovery and materials science.

Structure-aware scheduling and discrete tokenization may inspire future work on graph diffusion models.

Weaknesses

The improvements over strong baselines are incremental.

The empirical scope remains limited to standard benchmarks without broader downstream evaluation (e.g., property optimization, novelty, synthetic accessibility).

The impact appears specialized to molecular graph generation rather than broadly applicable across machine learning domains.

Originality

Strengths

The combination of RRWP-based structure-aware noise scheduling with VQ-VAE tokenization in discrete graph diffusion is technically interesting.

The node/edge-specific adaptive forward process is implemented in a detailed and reproducible manner.

Weaknesses

The overall contribution largely combines existing ideas (learnable forward diffusion, structural encodings, VQ tokenization).

The conceptual novelty lies more in integration and engineering than in fundamentally new theoretical development.

---

### Official Review · Reviewer_XeAn · 2026-03-08

**Soundness:** 2
**Presentation:** 1
**Significance:** 2
**Originality:** 2
**Overall Recommendation:** 2
**Confidence:** 4

**Summary:**

This paper suggests VQ-SAD, a discrete molecule diffusion model on a quantized latent space with better design choices over the baselines. To address the limitation that one-hot representations fail to capture the molecule’s symbolic information, the paper introduces a VQ-VAE to construct a discrete latent space for atom and bond codes. On top of that, to resolve the state-clashing problem, where distinct molecules collapse into similar states during forward process, the paper uses RRWP-based structural embeddings and learnable node/edge noise schedulers on structural information of individual nodes and edges rather than only on their types. On QM9 and ZINC250k datasets, VQ-SAD showed better distribution modeling ability compared to previous baselines.

**Compliance With Llm Reviewing Policy:**

Affirmed.

**Key Questions For Authors:**

- Since the introduced VQ-VAE is trained only for reconstruction without any additional external representation learning objective, can we actually justify that the obtained quantized latent features are more interpretable or advantageous for the performance of the generative model?

**Limitations:**

yes

**Strengths And Weaknesses:**

**Strength**
- The authors pointed out the limitations of the naive one-hot representation used in prior works, and raised the necessity for a latent space that contains more refined and enriched information.
- VQ-SAD achieves improved performance over existing baselines with better model designs of RRWP embedding and the replacement forward process.
---

**Weakness**

Presentation:
- The overall figures(for both methodology and results) show insufficient quality, and many of them even appear to be AI-generated, which questions the integrity of this work.
  - Incorrect format using the last year ICML template, with author affiliation error
  - Figure 1 is plotting unspecified scores without explanation.
  - Figure 2 contains unfeasible molecule structures that doesn't match the structural formula(+mixed color coding).
  - Figure 5 has diagrams with content overflowing its boundary.
  - Figure 6~11 also contains unfeasible/incomplete molecules with inconsistent color coding of atoms and bonds. Some of them(Figure 9, 11) assigned the incorrect numbering order to the molecules.

Soundness:
- The comparison baselines used in the benchmark results are limited; there exist several continuous unconditional/conditional diffusion models trained on the same dataset[1,2,3]. Since the changes of VQ-SAD over existing discrete diffusion models are modifications from which performance improvements can be relatively easily anticipated, a superior performance over a broad range of model families is required to emphasize the paper’s significance.
- Regarding the conditional generation results (Table 3), no metric is provided to show how close the generated molecules are to the actual input condition values.

---

[1] Equivariant Diffusion for Molecule Generation in 3D, ICML 2022

[2] Geometric Latent Diffusion Models for 3D Molecule Generation, ICML 2023

[3] Graph generation with diffusion mixture, ICML 2024

---

### Official Review · Reviewer_ucHL · 2026-03-11

**Soundness:** 2
**Presentation:** 2
**Significance:** 2
**Originality:** 3
**Overall Recommendation:** 2
**Confidence:** 3

**Summary:**

The authors propose a neuro-symbolic diffusion model for molecule generation named VQ-SAD. The paper first introduces a neural diffusion model, SAD (Structure Aware Diffusion), which utilizes Relative Random Walk Probabilities (RRWP) to learn an adaptive noise scheduling to prevent state clashing. SAD is then extended into VQ-SAD, where a frozen VQ-VAE acts as a fixed tokenizer and decoder, providing a more balanced and context-aware representation of molecular substructures compared to standard one-hot encodings. Both models are evaluated on the QM9 and ZINC250k datasets, showing small improvements over DiGress and MELD.

**Compliance With Llm Reviewing Policy:**

Affirmed.

**Final Justification:**

I recommend Rejecting this paper. While the motivation for using structure-aware embeddings and VQ-VAE codebooks to address state clashing and chemical context is well-reasoned and the methodology is clearly presented, the empirical evaluation is insufficient. Benchmarking is restricted to QM9 and ZINC250k, omitting larger datasets like Guacamol and recent SOTA baselines such as DeFoG and DisCo. The resulting performance gains over DiGress and MELD are marginal, making the model's significance unclear. Furthermore, the absence of ablations on diffusion steps and the lack of evidence isolating the VQ component’s independent impact leave the technical contribution unverified. As the authors did not submit a rebuttal to address these critical concerns regarding baseline comparison and modularity, my original assessment is reinforced.

**Key Questions For Authors:**

1. **Evaluating VQ Component Impact**: Could VQ-VAE tokenization be integrated into existing models like DiGress? Providing results for such a baseline would help isolate whether the performance boost comes specifically from the VQ-latent space or the SAD component.


2. **Collision Rate Evaluation**: In Table 4, you compare the collision rate of VQ-SAD against MELD. What is the collision rate for the SAD model alone? Since RRWP was introduced to reduce state-clashing, it seems more relevant to know if SAD solves this without the VQ component.


3. **Ablation Study on Steps**: How many diffusion steps ($T$) were used for the reported results? Have you experimented with varying the number of steps to see how it affects performance? How does the number of steps and running time of VQ-SAD compare against baselines?


4. **Learned VQ codebook**: Beyond the visual example in Figure 2, do you have quantitative evidence on what specific representations the VQ-VAE is learning?

**Limitations:**

Yes

**Strengths And Weaknesses:**

**S1. Motivation for SAD and VQ-SAD** :  The authors provide a good justification for their approach. They discuss the state clashing problem and justify using structure-aware embeddings for the noise schedule in SAD. Moreover, use of VQ-VAE codebooks is meant to address the inherent sparsity and imbalance of atom and bond types. In Figure 2, authors argue these codes provide better descriptors than one-hots by capturing the chemical context.


**S2. Methodological Details**: The paper provides a detailed explanation of the method, supported by algorithmic pseudo-code and diagrams.


**S3. Component-Wise Evaluation**: The authors evaluate SAD and VQ-SAD separately which helps in understanding the incremental impact of structural aware embeddings with and without the VQ-based tokenization.


**W1. Limited Evaluation and Baselines**: The experiments are restricted to only two datasets, QM9 and ZINC250k. Benchmarking on other datasets like Guacamol or MOSES would provide a clearer picture of the model's scalability and diversity. Furthermore, the baseline comparison is narrow and missing recent competitive methods such as DeFoG [1] and DisCo [2].


**W2. Sub-optimal Performance**: The performance gains over DiGress and MELD are quite small. Critically, more recent models like DeFoG [1] appear to achieve superior results on these same benchmarks, making the current performance of VQ-SAD less compelling. Maybe integrating the SAD and VQ components into a stronger base model would allow it to surpass current SOTA results.


**W3. Lack of Ablation on Diffusion Steps**: There is a notable absence of experiments regarding the number of diffusion steps ($T$). It is unclear how the model performs under different sampling strategies.


**W4. Ambiguous Contribution of Independent Components**: It is not entirely clear how the VQ component is working independently of the SAD structural embeddings. It might be worth testing the VQ component without SAD (e.g., adding it to DiGress) to isolate its impact.


[1] Qin et al., *DeFoG: Discrete Flow Matching for Graph Generation*, 2025.

[2] Xu et al., *Discrete-state Continuous-time Diffusion for Graph Generation*, 2024.

---

### Official Review · Reviewer_HxYm · 2026-03-13

**Soundness:** 2
**Presentation:** 1
**Significance:** 2
**Originality:** 2
**Overall Recommendation:** 2
**Confidence:** 4

**Summary:**

This paper proposes VQ-SAD, a two-stage approach for molecular graph generation. First, a VQ-VAE is trained to learn discrete codebooks for atom and bond types, mapping one-hot atom/bond representations to a larger set of learned codes that capture local structural context. Second, the frozen VQ-VAE codebooks serve as tokenizers for a discrete diffusion model with a learnable forward process. The diffusion model uses relative random walk probabilities as structural features to condition the noise scheduling. The method is evaluated on QM9 and ZINC250k, showing marginal improvements over DiGress and MELD baselines.

**Compliance With Llm Reviewing Policy:**

Affirmed.

**Key Questions For Authors:**

1. What specific chemical motifs or contexts do the learned VQ-VAE codes correspond to? The paper claims the codes capture "symbolic" information but provides no analysis of what the codebook actually learns.

2. Why does SAD perform substantially worse than MELD on conditional generation for dipole moment (Table 3)? This contradicts the claimed benefits of structural conditioning.

3. How does VQ-SAD compare to PARD, GraphDiT, and other recent state-of-the-art methods?

**Limitations:**

The authors mention the limitation of conditional generation reducing diversity. However, much more fundamental limitations are not discussed: the lack of comprehensive baselines, the small evaluation set, the absence of error bars, and the marginal improvements.

**Strengths And Weaknesses:**

**Strengths:**

- *Originality:* The idea of using VQ-VAE codebooks as a richer tokenization for molecular graph diffusion is conceptually interesting. The argument that one-hot atom types are informationally lossy and that learned codes can capture structural context has merit.

- *Significance (potential):* Addressing class imbalance in atom/bond types through codebook rebalancing is a practical concern in molecular generation.

**Weaknesses:**

- *Soundness:* The experimental evaluation is insufficient. Only four methods are compared and the improvements are marginal. Many important baselines are missing from the comparison.

- *Soundness:* The claim that VQ-SAD is "neuro-symbolic" is inflated. The codes do not correspond to interpretable chemical concepts in any verified way.

- *Presentation:* The paper has significant writing quality issues. There are grammatical errors throughout and inconsistent notation.

- *Originality:* Using VQ-VAE for molecular representation is not new (VQGraph, MoleBERT are cited). The combination with diffusion is straightforward. The RRWP-conditioned scheduling adds minimal novelty over MELD.

---

### Decision · Program_Chairs · 2026-04-30

**Decision:**

Reject

**Comment:**

The reviewers are all negative about the paper with strong concerns.
No rebuttal was submitted to tackle the concerns.